# NikshayChain: A Blockchain-Based Proposal for Tuberculosis Data Management in India

Madhuri Hiwale [1], Vijayakumar Varadarajan [2,3,4,*], Rahee Walambe [1,5] and Ketan Kotecha [1,5,*]

1   Symbiosis Institute of Technology, Symbiosis International (Deemed University), Pune 412115, India
2   School of Computer Science and Engineering, The University of New South Wales,
    Sidney, NSW 2052, Australia
3   School of NUOVOS, Ajeenkya DY Patil University, Pune 412105, India
4   Swiss School of Business and Management, SSBM Geneva, 1213 Geneva, Switzerland
5   Symbiosis Centre for Applied Artificial Intelligence, Symbiosis International (Deemed University),
    Pune 412115, India
*   Correspondence: vijayakumar.varadarajan@gmail.com (V.V.); director@sitpune.edu.in (K.K.)

**Abstract:** A recent development in the Internet of Things (IoT) has accelerated the application of IoT-based solutions in healthcare. Next-Gen networks and IoT, supported by the development of technologies such as Artificial Intelligence (AI) and blockchain, have propelled the growth of e-health applications. However, there are some unique challenges in the widespread acceptance of IoT in healthcare. Safe storage, transfer, authorized access control, and the privacy and security aspects of patient data management are crucial barriers to the widespread adoption of IoT in healthcare. This makes it necessary to identify current issues in the various health data management systems to develop novel healthcare solutions. As a case study, this work considers a scheme launched by the Government of India for tuberculosis care called Nikshay Poshan Yojana (NPY). It is a web-based Direct Benefit Transfer scheme to provide a nutritional incentive of INR 500/- per month to all tuberculosis patients. The main objective of this work is to identify the current implementation challenges of the NPY scheme from patient and healthcare stakeholder perspectives and proposes a blockchain-based architecture called NikshayChain for sharing patient medical reports and bank details among several healthcare stakeholders within or across Indian cities. The proposed architecture accelerates healthcare stakeholder productivity by reducing workload and overall costs while ensuring effective data management. This architecture can significantly improve medical care, incentive transfer, and data verification, propelling the use of e-health applications.

**Keywords:** blockchain; privacy; healthcare; data management; tuberculosis; Nikshay Poshan Yojana

## 1. Introduction

The availability of powerful and cheap computational power coupled with portable and accurate sensors has revolutionized how environmental and physiological data is collected in the big data era. This has also led to the foundation of Internet-of-Things (IoT)-based solutions. When combined with the ever-increasing capabilities of machine learning algorithms, IoT-based solutions have great potential to transform all aspects of human life. One such envisaged concept is smart healthcare, where IoT integration will help enhance citizens' overall quality and standard of living [1]. However, there are a few important challenges to such solutions; sharing patient medical and personal data with privacy and security guarantee are critical barriers to the widespread adoption of IoT in healthcare [2]. Therefore, it is necessary for any smart healthcare proposal to provide fundamental security standards such as integrity, authentication, trust, and authorized access in organizing, collecting, and sharing patient data [3]. Thus, it is essential to incorporate relevant technologies with IoT to design trustworthy healthcare data management systems.

In healthcare, care delivery models are usually organized around diseases such as HIV-AIDS, diabetes, cancer, and tuberculosis. For such disease conditions, national-level programs are implemented to provide care effectively. The primary purpose of such public health programs is to provide comprehensive disease-specific care and entire treatment services nationwide. The strict monitoring and successful deployment of these programs at every level are essential to achieve overall care quality. For an effective public health system, trustworthy coordination among several stakeholders requires transferring and organizing patient data in a secure, trusted, and privacy-preserving manner. In India, various national-level health programs are implemented to strengthen public health and improve health outcomes. This work focuses on one of the national-level schemes called Nikshay Poshan Yojana (NPY) for Tuberculosis (TB) patients launched by the Government of India.

### 1.1. Related Works and Main Contributions

Currently, healthcare institutes around the world are severely affected due to the COVID-19 pandemic [4]. The current global COVID-19 crisis has brought back focus on the existing and ever-evolving infectious diseases to ensure a healthy life for all. Thus, this pandemic has forced the healthcare sector to focus on the issues related to controlling contagious diseases. The world has been suffering from a few contagious diseases since ancient times. One such important disease is tuberculosis. To effectively manage this ancient infectious disease, there is a need to understand the current progress and issues with the national-level TB management systems.

Tuberculosis is an airborne contagious infectious disease representing a considerable global public health burden [5]. According to the Global TB report, TB prevention and care progress are very slow [6]. As per the World Health Organization (WHO) report, more than 10 million people are suffering from tuberculosis around the globe [7]. Despite significant improvements, India remains among the highest TB burden countries. In India alone, almost 2.6 million people suffer from TB, approximately 26% of the global burden, with more than 1000 deaths every day due to TB [8].

There are several risk factors for TB disease. According to a global report, undernutrition, alcoholism, smoking, diabetes, and HIV, are well-known risk factors [9]. In 2013, the WHO reported that undernutrition increases TB risk, leading to malnutrition. In developing countries, undernutrition and malnutrition lead to disrupting of the immune system and reactivating TB infection [10]. In India, there is a strong association between TB and malnutrition; as most TB patients are either poor or weak, managing malnutrition is vital for TB patients. Thus, nutritional support is essential to deal with TB [8]. To cure TB disease, routine follow-up for the entire duration of treatment and a nutritious diet are important factors. In India, the integration of nutritional support policies plays a crucial role in eliminating TB [11]. The latest global TB report has also highlighted that the patients do not participate actively in the current TB elimination process and are typically unaware of how their data is stored, accessed, and used by stakeholders. Therefore, there is a need for a patient-centric trustworthy approach to increase patient participation in TB treatment.

Considering this enormous burden, the Government of India has launched various schemes to improve the overall quality of TB care in India. As per the WHO recommendation, the Government of India, under the "National strategic plan for Tuberculosis 2017–2025", launched a scheme called Nikshay Poshan Yojana in April 2018 to provide nutritional support for TB patients. The word Nikshay is formed by combing two Hindi words; Ni and Kshay mean end tuberculosis. The NPY is a Direct Benefit Transfer (DBT) scheme to provide a nutritional incentive of INR 500/- per month to all notified TB patients for the entire duration of treatment. Various stakeholders and steps are involved in carrying out this process smoothly, from patient notification to the NIKSHAY (a web-based tuberculosis patient management platform in India) portal to transferring a nutritional incentive of INR 500/- per month till the end of treatment. The patients can utilize the

incentive to fulfill their dietary needs or towards treatment-associated expenses that help to enhance overall TB treatment outcomes.

Despite the significant benefits, there are still a few issues and operational challenges associated with this TB scheme and its proper implementation. In India, to ensure the successful execution of the NPY scheme, leading to patients receiving full advantage of benefits, there is a need to understand the current workflow and the technology-related implementation challenges this novel scheme faces. For this reason, this work identifies all technical challenges from the patient and the stakeholder perspectives.

In the current workflow, the main hurdles with the NPY system are the multi-layer approval procedure, multiple layers of data verification, the extra workload burden to perform validation, a time-consuming format, a lack of patient-centric approach, and trust for sharing patient data, delay in receiving the incentive, and data duplication [12]. There is an urgent need to bring together all the healthcare stakeholders, government, and other technology innovators to develop an advanced technology-based solution to deploy and strengthen the NPY scheme successfully.

Blockchain has the potential to deal with data management issues and successfully deploy the NPY system. Blockchain is a distributed ledger consisting of a chain of blocks linked together in an immutable and secure manner [13]. Blockchain facilitates smart contracts logic, a digital agreement between two parties without third-party involvement [14]. Blockchain is an excellent solution for offering the desired level of trust, transparency, and decentralization in entire healthcare systems, thereby enhancing privacy and security. Blockchain-based architectures can minimize overall security monitoring costs and offer protection against attackers trying to obtain private information.

In this work, we have introduced a novel blockchain-based architecture called Nikshay-Chain to develop a patient-centric, effective data-sharing network among all the stakeholders involved in the NPY process. NikshayChain is a trustworthy Hyperledger fabric-based architecture. It will help build a robust patient-centric TB system for effective data management, contributing to smart healthcare. With smart contracts, data verification and incentive transfer can occur automatically and efficiently, reducing healthcare stakeholder workload and developing smooth coordination between patients and stakeholders [15]. With its inbuilt smart contract logic, blockchain will accelerate the productivity of all the healthcare stakeholders involved in the NPY system and help enhance overall care quality [4].

However, very few researchers discuss the use of blockchain in TB surveillance and implementing TB management systems such as NPY. It is still in the initial development phase. In this regard, this paper attempts to emphasize the integration of blockchain in the NPY scheme to solve the current implementation issues. Hence, the first objective of this work is to understand the current status, working, and data management challenges of NPY from patient and health stakeholder perspectives. The second objective is to introduce a blockchain-based solution to solve the issues associated with the smooth implementation of the NPY scheme.

In summary, the main contribution of this work is:

- Discussion and analysis of the working of Nikshay Poshan Yojana, the responsibilities and challenges faced by different stakeholders, and the data management issues associated with this novel scheme.
- Proposing a method based on blockchain for smart healthcare and understanding how a blockchain-based architecture can help build patient-centric and efficient data-sharing with a specific example of the TB scheme.
- Proposing the architecture for implementing the blockchain-based model for the NPY scheme, which can be extended to any public health scheme.

### 1.2. The Paper Organization

The rest of the paper is organized as follows: The materials and method section describes the methods used to find the relevant information and literature. The third section first describes the current working and challenges present in the NPY system

and emphasizes the role and responsibilities of different stakeholders involved in the system. Secondly, it highlights the applicability of blockchain technology to the healthcare scenario and proposes a blockchain-based architecture by integrating blockchain into the NPY scheme. In the discussion part, we highlight the potential benefits of blockchain that considerably impact removing the barriers and challenges present in the current NPY scheme. Conclusively, this work intends to develop a permissioned blockchain network that allows secure and automated data verification. Figure 1 depicts the overall structure of the manuscript.

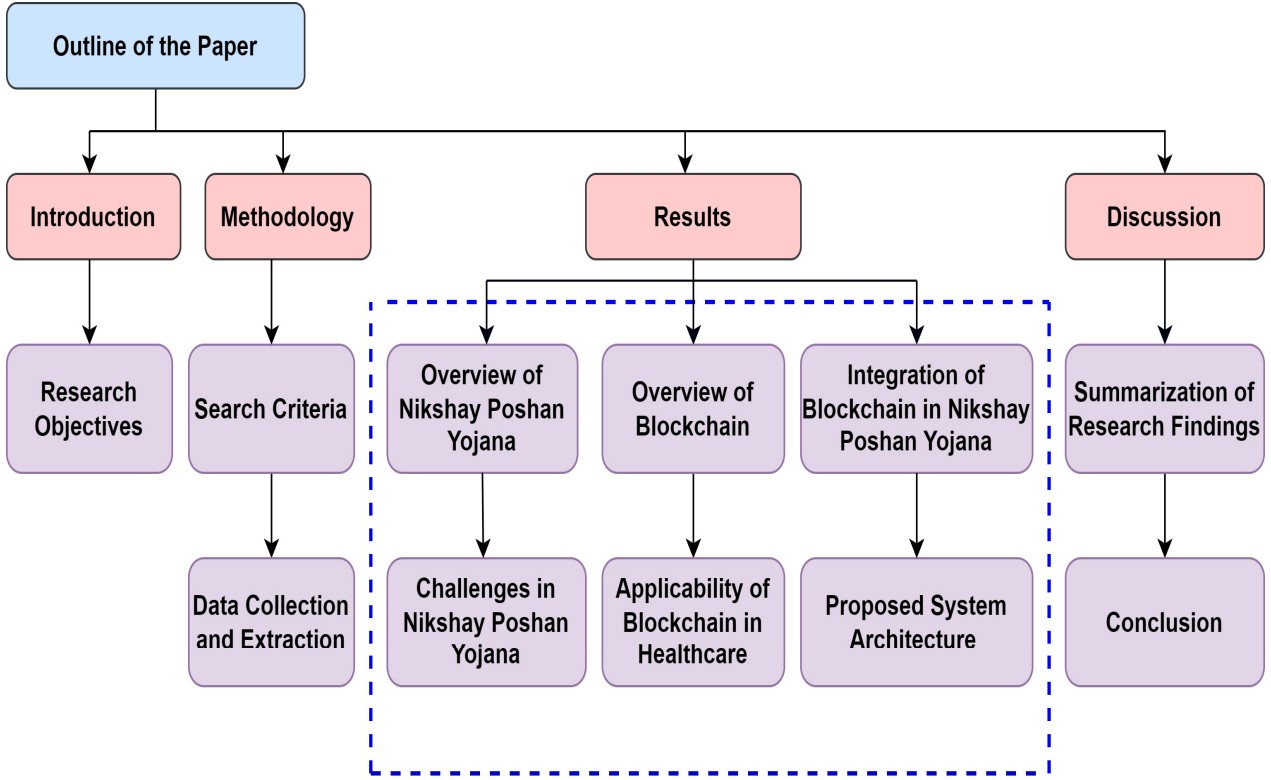

**Figure 1.** Outline of the paper.

## 2. Materials and Methods

Based on objectives, we gathered all the relevant information from the official websites of Nikshay Poshan Yojana, Central TB Division, and WHO annual TB report to know the status of TB in India and the current state-of-the-art working of the NPY scheme. We have searched scientific databases such as PubMed and Scopus to find relevant literature.

### 2.1. Search Criteria

The objective-specific search queries such as "Tuberculosis and India and Challenges," "Nikshay Poshan Yojana," and "Tuberculosis and Stakeholders and India" were executed in respective databases to fetch the relevant literature. These search queries help find the current status of TB in India and understand the challenges and gaps responsible for eliminating TB from India. Blockchain-specific search queries were executed to understand the blockchain working, such as "Blockchain and Healthcare and Data sharing" and "Tuberculosis and Blockchain." These search queries helped us understand blockchain's role in health data sharing.

### 2.2. Data Collection and Extraction

The data collection process for this study was conducted in mid-2022. We have searched the official websites of the Central TB Division, WHO, and Nikshay Poshan Yojana to collect data useful for the analysis. The last five-year conference papers and

journal articles written in English from Scopus and PubMed data sources were considered for further literature survey. This study excluded the documents which were not relevant to address formulated objectives. The papers were analyzed manually by screening the title and abstracts of the articles. Finally, by filtering the title and abstracts and removing the duplicate studies, the studies that focused on formulated objectives, such as NPY scheme implementation challenges and blockchain, were included for in-depth analysis.

### 3. The Proposed Blockchain-Based NPY System

Firstly, this section describes an overview of the NPY system and data management issues present in the current system from the perspective of the patient and healthcare stakeholders. Second, it represents the applicability of blockchain and its benefits for the healthcare scenario. Finally, this section includes the proposed blockchain-based system architecture.

### 3.1. Nikshay Poshan Yojana (NPY)

The Nikshay Poshan Yojana is a centrally sponsored scheme under the National Health Mission (NHM). The NPY is an incentive-based Direct Benefit Transfer (DBT) scheme [16]. The NPY scheme is implemented across all the states in India. This TB-specific scheme includes all the notified and registered TB patients from the public and private sectors on the NIKSHAY web-based portal. TB takes time to cure; the usual course of treatment lasts from 6 months to 24 months. The main goal of this scheme is to provide INR 500/- to all reported TB patients per month for the entire duration of treatment. Each TB patient from the public or private sector must register on the NIKSHAY platform to avail of these benefits. The patient will have to come to nearby TB treatment centers with a copy of the Aadhaar (National identification number for Indian citizens), bank details linked with the Aadhaar card, and medical certificates for TB treatment as proof for claiming the benefits under the NPY. The patient submits an online application form which is stored in the database. If the notified patient fulfills all the criteria, he can get the benefit of INR 500/- each month, which gets credited to his bank account and continues until TB gets cured. This DBT scheme is incorporated with a centrally sponsored Public Financial Management System (PFMS) [12]. The PFMS is a web-based application responsible for transferring all the funds to the beneficiary under the schemes launched by the Government of India.

The NPY provides financial assistance to all registered TB patients that help boost their health by improving their nutritional status. Cash transfer improves notification and surveillance, enabling healthcare stakeholders to ensure treatment completion and quality of care [17]. The NPY encourages poor patients to continue their treatment and helps them to fight against malnutrition.

#### 3.1.1. Stakeholders Involved in NPY Scheme

Various stakeholders and steps are involved in carrying out this process smoothly, from patient notification to the NIKSHAY portal to transferring INR 500/- per month till the end of treatment. Each stakeholder has responsibility, from the verification of documents to the transfer of incentives to the registered bank account of a patient. Stakeholders are categorized into three levels: maker, checker, and approver. The maker is responsible for entering data on the NIKSHAY portal, maintaining hard copies of all the patient documents, making a unique beneficiary list, and sending it to further approval. The checker is responsible for verifying patient records and then sending these records for approval to the next level. The approver is responsible for validating the beneficiary list and sending the list of the approved beneficiary to the district health society (DHS) [16]. DHS transfer the cash to the bank account of the authorized beneficiary through the Public Financial Management System (PFMS) [18] portal. Figure 2 depicts the task performed by each stakeholder and the steps involved in the NPY system.

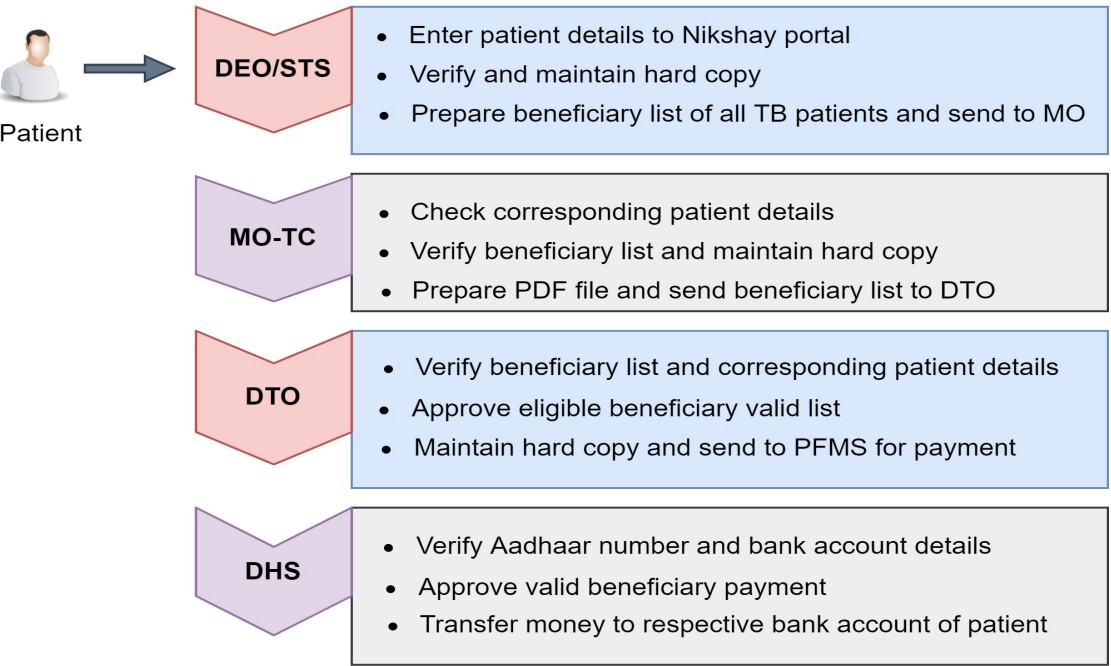

**Figure 2.** Steps and stakeholders involved in the NPY.

3.1.2. Role and Responsibility of Stakeholders

- Data Entry Operator (DEO):

The data entry operator is mainly responsible for entering TB patient data from the private or public sector into the NIKSHAY platform. DEO Updates necessary information on treatment outcomes and follow-ups in the NIKSHAY platform. DEO also updates and validates the Aadhaar card number and bank details of beneficiaries in the NIKSHAY and maintains the hard copies of the documents. The health staff facilitates the beneficiary opening of bank accounts, helps with Aadhaar enrolment at the nearest Aadhaar enrolment center, and keeps the local Aadhaar enrolment centers list.

- Senior TB Supervisor (STS):

STS ensures notification of all TB patients in NIKSHAY with the Aadhaar number, complete address, mobile number, and bank details. STS and DEO prepare a list of beneficiaries in NIKSHAY on the first day of every month. They verify and validate Aadhaar based on information furnished by the beneficiary and submit a validated beneficiary list to the medical officer of TB control.

- Medical Officer-TB Control (MO-TC):

MO-TC ensures timely submission of checklist of beneficiaries by STS / Health Staff through DEO, check-in, particularly for mobile number, Aadhaar number, bank account number, and IFSC (Indian Financial System Code) code of the beneficiary bank account. They also check for duplication of patient detail. The MO-TC then submits the validated list to the District TB officer for further processing.

- District TB Officer (DTO):

The DTO ensures that all the district MO-TCs provide the validated beneficiaries list on time. The DTO is responsible for training the health staff on DBT and ensuring regular digital payment using DBT by the Public Financial Management System.

- State TB Officer (STO):

The State TB Officer is mainly responsible for supervising the district-wise progress of updating of Aadhaar card and bank details of beneficiaries in the NIKSHAY portal. STO also reviews, plans, and ensures funds for financial support to all TB patients, private

providers, and treatment supporters. Finally, STO checks the progress of the transaction of financial incentive through DBT.

### 3.2. Challenges in the NPY Scheme

Since the release of this novel scheme, few operational challenges have been posed against its proper implementation. According to the current India Tuberculosis report (2022), only 62 percent of total notified cases around the country received at least one nutritional incentive in 2021. In the states such as Delhi, Maharashtra, and Punjab, the percentage of receiving incentive at least once in 2021 is less than 50 percent [19]. There is a need to identify the challenges urgently to improve the outcome of the NPY. Figure 3 shows the current challenges present in the NPY scheme from patient perspective, stakeholders perspective and technology perspectives.

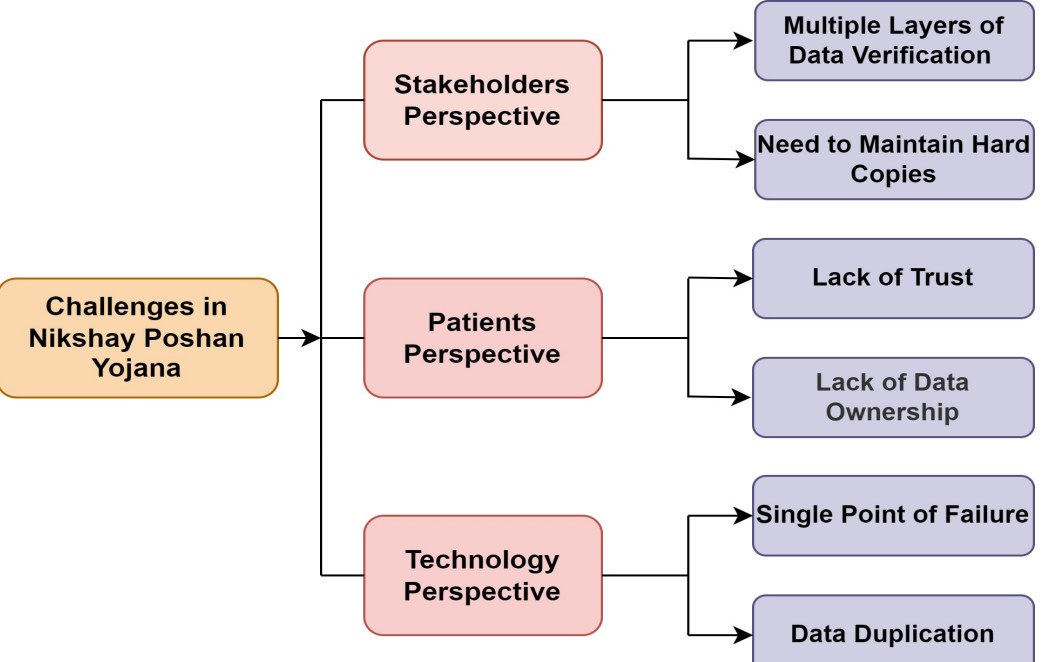

**Figure 3.** Challenges in the Nikshay Poshan Yojana.

This scheme improves the notification of TB patients in general, but still, some urgent issues need to be solved to take full advantage of this novel scheme. From the stakeholder perspective, the multistep approval procedure, multiple layers of data verification, the extra workload to perform validation, repetitive tasks, and maintaining the hard copies of documents at every level lead to delays in receiving the incentive. From the patient perspective, lack of trust and transparency in sharing personal data by patients with government schemes and a lack of data ownership. A few other issues, such as time-consuming and complex formats, data duplication on the NIKSHAY platform, and unlinked Aadhaar and bank account details of patients, are associated with this scheme [20].

According to A S. Nirgude et al., the critical challenge noted by the health stakeholders is related to transferring incentive [20]. They found that incentive transfer is a complex and time-consuming process with multiple layers of approval and involves too much paperwork. This complex procedure causes a delay in receiving incentive. In this scheme, additional paper-based documentation is required at every level, increasing the extra burden on the existing documentation workload and causing a delay in cash transfer. It thus fails to achieve the primary goal of this scheme.

Dr. Kumar R et al. surveyed the implementation of this scheme [12]. They noticed none of the patients had received the incentive on time for the entire course of the treatment. According to health stakeholders, the main challenge is the increased workload and com-

plex reporting format of the DBT scheme; they note this as the main hurdle in the current implementation of the scheme. Health providers must maintain the paper documentation and repeat the same process monthly. Along with this linking a bank account with the Aadhaar number is a hurdle in receiving the incentive.

Patel BH et al. discussed various challenges in implementing the DBT cash transfer scheme [18]. They classified challenges broadly as system-related, patient-related, and provider-related. In the private sector, patients lack the trust to share their details, such as bank account and Aadhaar number. Patients are concerned about the security of their data. Migrant patients cannot open bank accounts due to the lack of valid residence/identity proof. Patients are not able to receive an incentive on time. An error in the linkage between the NIKSHAY portal and the Public Financial Management System (PFMS) portal increases the work burden on health providers because they have to manually prepare the beneficiary list from the PFMS portal, causing a delay in the transfer of incentives.

According to Thakur et al. [21], In India, there is an urgent need to improve the efficiency of the current public and private health infrastructure. There is a need to focus on a few non-health factors, like providing more comprehensive access to healthcare, increasing awareness and resources, and adequately managing infrastructure and human resources. Other factors like improving notification and adopting advanced technologies in TB services are beneficial to strengthen the TB elimination program. Similarly, Sachdeva [22] mentioned integrating modern technologies like data mining and Artificial Intelligence in TB surveillance to accelerate the TB elimination progress.

Overall, the NPY scheme is very complicated and time-consuming, as all the stakeholders have to perform repetitive tasks and paperwork every month. Patient reluctance to share sensitive details and lack of coordination between patients and healthcare providers are other challenges associated with this scheme. In this regard, emerging blockchain technology is feasible with its fascinating features to deal with the abovementioned issues. The following section describes the key benefits of blockchain for the healthcare sector.

### 3.3. Overview of Blockchain in Healthcare Scenario

Blockchain can be defined as a distributed and immutable data structure. Since the invention of Bitcoin (a cryptocurrency) in 2008 [23], the growth and popularity of blockchain are achieving new heights. With its inherited characteristics, it became suitable technology for solving many issues in the healthcare system [24]. Each block contains its hash, the hash of the previous block, and several transactions. Figure 4 shows the structure of blocks in the blockchain.

Public, permissioned, and hybrid blockchains are well-known types of blockchain networks [25]. In a public network, data is accessible to all participating nodes, e.g., Bitcoin. In a permissioned network, information is accessible only to authorized nodes that have permission, e.g., Hyperledger Fabric. A hybrid network combines public and permissioned networks, e.g., the Dragon chain [26]. The appropriate blockchain platform is chosen based on the application requirement.

Key Features of Blockchain

- Decentralize: Most healthcare systems rely on centralized repositories to store sensitive medical data. The main issue with a centralized repository is a single point of failure that can destroy the entire network. In such cases, the decentralized nature of blockchain helps overcome the problem of a single point of failure.
- Immutable: When a transaction is validated and recorded on the blockchain, it becomes impossible to change this validated data. Thus, it helps to record immutable agreements between various untrusted nodes and enhances business growth, ensuring data quality and integrity.
- Trust and Transparency: Blockchain network is trustworthy. It uses consensus mechanisms and cryptography techniques to store data in the network. The consent of the majority of participating nodes is needed to keep data in the network. Blockchain

develops transparency and trust among participating healthcare stakeholders to share patient-sensitive medical data with these two mechanisms.

- Eliminating Middleman: Blockchain has great potential to overcome the need for intermediaries. Smart contracts enable different stakeholders to communicate and transact directly on a peer-to-peer basis [27].

- Smart Contract Logic: It is one of the fascinating features of blockchain [28]. With smart contracts, blockchain overlaps traditional paper-based contracts and converts them into digital agreements [29]. Smart contracts provide network automation and enable codifying the digital agreement between untrusted parties without central supervision [30]. Smart contract verifiability and auto-enforcing ability facilitate the execution of predefined business rules in a distributed network, where each participating node is equal to predefined authority. Smart contract code, once executed on the blockchain, no one can modify or change that code. Multiple blockchain platforms support smart contract execution by using numerous programming languages. In Ethereum, solidity is a well-known language. In Hyperledger Fabric, Go, JavaScript, and Java are major programming languages. Regardless of blockchain type, smart contracts help to encode business logic. Smart contracts are self-executing and tamper-proof programming codes that are responsible for reshaping business processes. Smart contracts embedded in blockchain enable predefined conditions of the agreement to be executed automatically without any third-party intervention. As a result, it can reduce administration costs and improve the efficiency of the process [30].

With these exciting and unique features, blockchain facilitates advanced technology-based solutions to handle many issues in healthcare systems [25]. In a blockchain network, data is spread among all participating stakeholders. Each one holds an identical copy of data or ledger. There exist agreements that are consensus-based protocols to guarantee consistency of the ledger in multiple stakeholders. In the blockchain, the well-known approach that helps to reach consensus are Proof-of-Work (PoW), Proof-of-Stake (PoS), Practical Byzantine Fault Tolerance (PBFT), etc. Each consensus mechanism has its benefits and drawbacks [25]. Table 1 summarizes the relevant studies that implement blockchain-based architecture for healthcare applications.

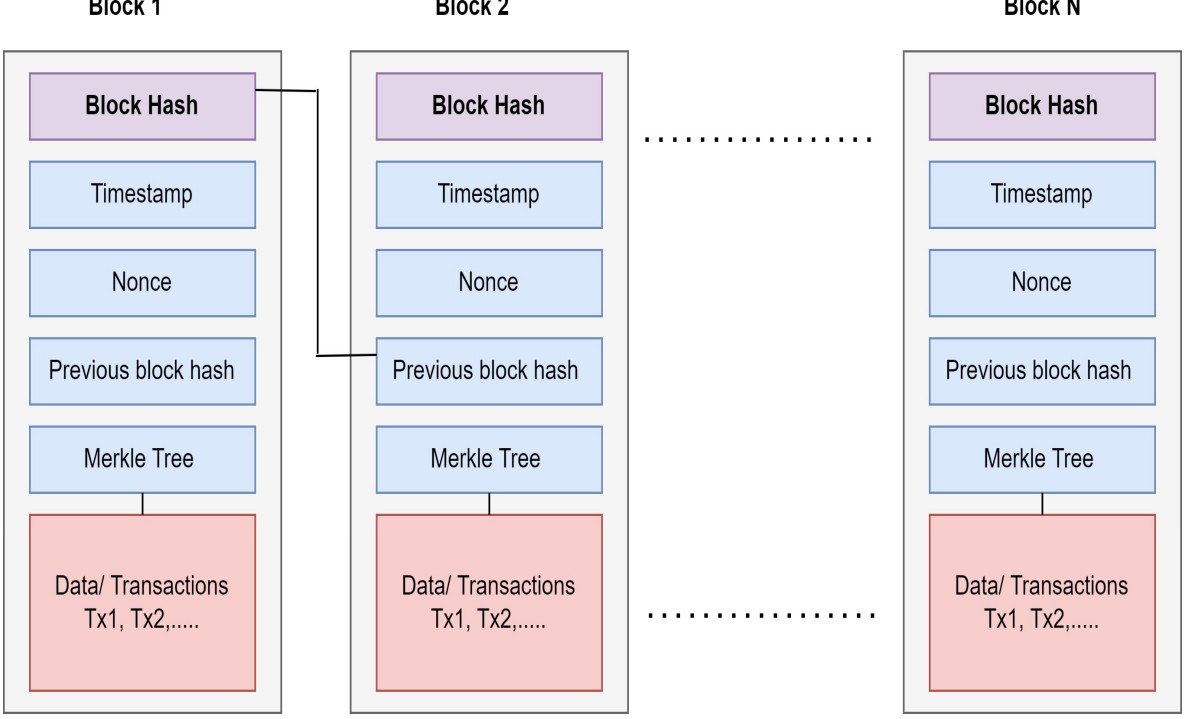

**Figure 4.** Structure of the blocks in Blockchain.

Blockchain has a significant impact on the healthcare domain [14]. It is a promising technology that provides the solution to store and retrieve secure and trustworthy health data among stakeholders. It enables tamper-proof and secure health data sharing through consensus mechanism and encryption without depending on any third party or centralized entity. Recently, several researchers have designed blockchain-based healthcare architecture to facilitate Electronic Health Records (EHR) management [31] and patient-centric data sharing between various stakeholders [32]. Some research studies have employed smart contract functionality to write down stakeholder and patient-centric agreements to create authorized data access among all participating members [33]. Few research studies have designed a complete decentralized healthcare model [34].

**Table 1.** Summarization of relevant studies using blockchain-based solutions for healthcare.

| Sr. No. | Authors | Key Findings | Limitation |
|---|---|---|---|
| 1 | S. Bhattacharya et al., 2019 [35] | With blockchain's help, the surveillance system can become more effective and faster than traditional surveillance in terms of coverage, durability, consensus, selective privacy, uniqueness, and timing. The blockchain avoids data duplication and improves patient-provider communication. | Lack of frameworks for implementation and regulation, concerns for cost-effectiveness and interoperability. |
| 2 | Vijay Kumar Chattu et al., 2019 [36] | This paper explores the basics of blockchain, its applications, quality of experience, and advantages in disease surveillance over the other widely used real-time and machine-learning techniques. The blockchain helps to achieve data integrity. | There is a need to address security and scalability issues. |
| 3 | Filipe Bernardi et al., 2019 [5] | This paper presents a blockchain-based TB network proposal to share tuberculosis data between several stakeholders. Due to permissioned blockchain networks, stakeholders guarantee security and auditability to transmit or receive data; they can manage their data access levels, users, and privacy. | Lack of scalability, interoperability |
| 4 | Cichosz et al., 2018 [32] | Authors propose using NEM Blockchain to develop patient-centered governance of health data, multi-signature contracts for access control of data management, and data encryption to allow privacy and control of health care data. | Lack of scalability |
| 5 | Jie Xu et al., 2019 [37] | The authors propose a health chain, a large-scale health data privacy-preserving scheme, and encrypted health data to conduct fine-grained access control in this paper. Privacy is maintained; users can effectively revoke or add authorized doctors by leveraging user transactions for key management. | Interoperability, lack of open standards, trust between all parties, and data integration |
| 6 | S. Tanwar et al., 2020 [38] | The authors propose a system architecture and algorithm for a patient-centered approach to providing an access control policy with symmetric key cryptography to a different healthcare provider. | There is a need to address the privacy issue |
| 7 | Peng Zhang et al., 2018 [39] | It enables scalable and secure clinical data sharing with established trusts, providing clinicians with secure and scalable collaborative care decision support. | High deployment cost, interoperability |
| 8 | Asma Khatoon et al., 2020 [40] | This paper proposes using a smart contract from the Ethereum platform to build record ownership, permission, data integrity, and patient-provider coordination. Smart contracts have been designed for different medical workflows and managing data access permission between various entities in the healthcare ecosystem. | The cost of deployment of the smart contracts is higher |
| 9 | Lee et al., 2020 [41] | This paper designed an Ethereum-based blockchain model for health data exchange to guarantee data availability and integrity. | Data Privacy |
| 10 | Rahman et al. [42] | The authors designed private blockchain-based architecture for cancer patients to provide secure data exchange among healthcare stakeholders. | Lack of Scalability |
| 11 | Dib et al. [43] | In this paper, the authors created a Hyperledger fabric blockchain-based model to develop a user-centric network for effective data sharing between patients and stakeholders. | Lack of Scalability |
| 12 | EI Mojdoubie et al. [44] | The authors proposed a blockchain-based smart healthcare model to share data in a privacy-preserving manner. | Interoperability |
| 13 | Yazdinejad et al. [31] | Blockchain creates a decentralized network for remote patient monitoring with valid access policies and secure data transmission. | Lack of Scalability |
| 14 | Singh et al. [45] | The authors present a novel blockchain-enabled patient-oriented framework for healthcare applications. This Hyperledger-based architecture provides a secure data-sharing platform for stakeholders of the healthcare domain. | - |

*3.4. NikshayChain: A Proposed Blockchain-Based NPY Architecture*

This subsection presents the proposed NikshayChain architecture based on the Hyperledger Fabric permissioned blockchain platform.

### 3.4.1. System Overview

In the current workflow, when the patient gets diagnosed with TB infection, he must be notified to the cloud-based NIKSHAY portal by the respective healthcare provider from any public/private healthcare sector where the patient is seeking TB treatment. When the patient demographic and medical test details are stored in the NIKSHAY portal, a unique NIKSHAY id is generated. Only the patient with a valid NIKSHAY id is eligible to enroll in the NPY scheme and receive a nutritional incentive of INR 500 Rs/. To avail of the benefits, the patient must provide sensitive details such as an Aadhaar number and bank details (IFSC code, bank account number). All these sensitive data get stored in the cloud-based NIKSHAY web portal. All the stakeholders access this web portal to approve and verify the patient details and maintain the documentation. Each stakeholder (DEO, STS, MO-TC, DTO, and STO) must perform monthly verification and approval and keep hard copies of documents. First, data is approved and verified by DEO then the same data again verify and approved by all the stakeholders. After final approval, the beneficiary list is sent to the PFMS portal to transfer the incentive through DBT.

### 3.4.2. A Proposed System Architecture

The Current workflow depends upon a centralized cloud-based portal that can lead single point of failure and a lack of transparency and trust among stakeholders. On the other hand, the proposed blockchain-based decentralized architecture offers smooth trustworthy coordination among all the stakeholders.

This work proposes permissioned blockchain-based architecture, the NikshayChain network, to share TB patient data with appropriate data access policies among several stakeholders. The architectural design of the proposed blockchain-based NikshayChain network for the NPY scheme is presented in Figure 5. The proposed architecture is built over the Hyperledger Fabric platform. Hyperledger Fabric is a permissioned blockchain platform that involves all stakeholders responsible for transferring the incentive through the NPY scheme. Each stakeholder has some predefined role and responsibilities. Based on the predefined responsibilities, smart contracts will get executed automatically and granted permission to verify the data. In the Hyperledger Fabric platform, a smart contract called chain code is responsible for implementing business logic based on the digital agreement among the stakeholders. To join this network, each stakeholder has an identity.

In Hyperledger Fabric, Certificate Authority (CA) tool is responsible for registering and enrolling all participating stakeholders and providing digital identity. Only authorized stakeholders such as patients, DEO, STS, MO-TC, DTO, and STO have permissioned to join the NikshayChain network.

In the proposed workflow, The DEO will act as admin and be responsible for adding all the participating stakeholders into the blockchain network. The patient has all control over his data; with a smart contract, the patient specifies the access permission before uploading data on the blockchain. Once the treatment is completed, the patient can deny access to his data.

Firstly, the patient diagnosed with TB and has Nikshay Id needs to approach the concerned DEO of the Tuberculosis unit. The DEO requests the patient to submit the documents required for incentive transfer, such as Nikshay Id, Aadhaar number, test result, and bank details. DEO verifies these documents with original documents. Depending upon a patient's TB test results, DEO mentions how many months the patient can receive the nutritional incentive. After data verification, DEO takes the patient consent, generates a unique patient ID, and broadcasts data into a permissioned blockchain network. The smart contract will get executed automatically before broadcasting the data into the network. With a smart contract, preconditions are checked, such as whether the patient ID and

Nikshay Id are unique, the bank details, Aadhaar number, and the number of months the incentive transfer is submitted or not. Once all these conditions are satisfied, data can be broadcasted into the NikshayChain network.

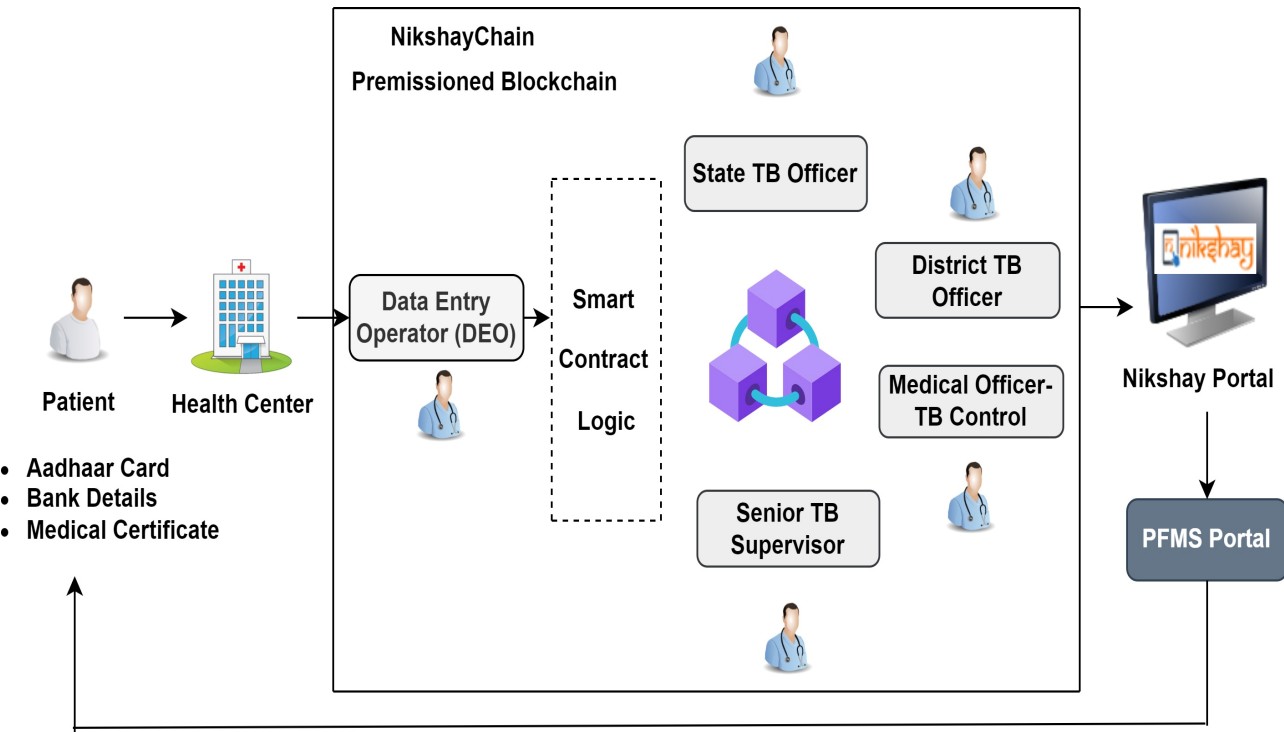

**Figure 5.** Proposed blockchain-based architecture for the NPY scheme.

All the stakeholders who are part of the network receive identical copies of the data, so there is no need to maintain hard copies or verify the same document repeatedly. Once the data is verified and approved by the DEO, there is no need to verify the same data by other stakeholders. They need to check two conditions (predefined in the smart contract); first, is the DEO approves the data or not, and second, is the patient submitted the test report and is eligible to receive an incentive for the current month. Once these conditions are satisfied, the patient is added to the valid beneficiary list. After approval, a valid beneficiary list is uploaded on the NIKSHAY portal. District Health Society (DHS) checks the correct list and makes payment advice to respective banks through the Public Financial Management System (PFMS). Finally, the patient receives the incentive in his bank account.

The blockchain makes the NPY process more straightforward, secure, and transparent. This framework helps reduce the extra workload burden on stakeholders as smart contracts automate data verification, and there is no need to maintain hard copies of documents. The patient controls his data, which develops trust and transparency in the system. The patient receives an incentive without any delay. The immutability feature of blockchain helps to avoid data duplication. This proposed architecture enables a cost-effective, robust network for TB patients.

## 4. Discussion

Smart healthcare is an important aspect of the concept of quality of life. A healthy society maintains the exact balance in every aspect of life. The demand for smart healthcare is increasing for society's betterment to provide quick and efficient healthcare services. The crucial goal of smart healthcare is to enhance the health treatment of the patient, boost the quality of patient life, reduce the workload burden on healthcare stakeholders, reduce cost, and create a secure collaboration between patients and healthcare stakeholders.

Healthcare integration with modern technologies such as the Internet of Things (IoT), AI, and blockchain is necessary to achieve this goal. Blockchain technology has a huge impact on healthcare as it enables to design of patient-centric architecture with decentralized and trustworthy collaboration among the stakeholders. In this regard, this work proposes a blockchain-based smart healthcare architecture to overcome the issues in the NPY scheme and boost the overall TB care quality in India.

The NPY is a novel scheme by the Government of India to tackle TB in India. However, there are a few critical issues with this scheme. If not corrected in time, these issues may lead to less than desired impact of this vital scheme. The main goal of the NPY-DBT procedure is that each TB patient should receive an incentive on time for the proper duration. This paper aimed to review challenges associated with the NPY-DBT strategy. To achieve this goal, we observed a need to create a network for TB that makes the transfer of incentive simpler, faster, and transparent and reduces the multiple layers of data verification among different stakeholders of the NPY process.

Based on a review of the challenges associated with the NPY scheme, we propose the use of blockchain technology to solve the associated problems. We presented the blockchain-based TB architecture called NikshayChain to automate data sharing among all the stakeholders. With this framework, the following advantages can be achieved by the NPY scheme.

- It will strengthen the overall national TB surveillance system in India.
- With the blockchain, the process of data verification and transfer of incentive takes place automatically and efficiently. The NPY process can become simpler, faster, and more transparent.
- It will improve service quality and will increase operational efficiency. It will eliminate the need for multiple layers of approver to reduce the delay in verification and transfer of incentive to the respective bank account of the patient.
- This framework can transfer incentive to a beneficiary on time, improving transparency, efficiency, and accountability in the TB surveillance system.
- It will reduce the overburden on all TB stakeholders and eventually boost the DBT process, and no need to maintain extensive paper-based documentation.
- Increase the trust among the patient to share their document, and provide personalized care to TB patients.

*Limitation*

This study has recognized some limitations. For this work, we have only analyzed web-based information and the research documents published in PubMed and Scopus data repositories, resulting in missing valuable documents relevant to the formulated objectives. As mentioned, we have only focused on conference papers and journal articles, which may miss some relevant information published as grey literature.

## 5. Conclusions

Next-generation networks and effective data management, security, and privacy techniques have propelled the development and use of e-health applications. This work demonstrated the use of blockchain technology for TB data management systems. Blockchain is a flexible solution for the administrative and institutional culture of organizations. Blockchain can make the process of data validation and verification faster, establish trust, and facilitate transparency that enables accountability. Blockchain with smart contract logic removes the workload burden and promotes a patient-centric approach that facilitates smooth coordination between all the stakeholders. To effectively implement and manage such government-supported national schemes (in any part of the world), it is imperative to employ more innovative solutions and digital technologies such as blockchain to ensure effective data management. With an example of the NPY-DBT scheme for TB care, it is demonstrated that the integration of the blockchain helps in the development of a patient-centric reliable, data-sharing TB system. In the future development, there is a need to design

a completely robust, trustworthy, and patient-centric healthcare network for tuberculosis to achieve the goal of a TB-free India.

**Author Contributions:** Conceptualization, M.H., R.W. and K.K.; methodology, M.H.; validation, R.W., V.V. and K.K.; investigation, M.H. and K.K.; resources, K.K.; writing—original draft preparation, M.H.; writing—review and editing, M.H. and R.W.; supervision, R.W., V.V. and K.K. All authors have read and agreed to the published version of the manuscript.

**Funding:** This research received no external funding.

**Institutional Review Board Statement:** Not applicable.

**Informed Consent Statement:** Not applicable.

**Data Availability Statement:** Not applicable.

**Conflicts of Interest:** The authors declare no conflict of interest.

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
