# Peer review of "NikshayChain: A Blockchain-Based Proposal for Tuberculosis Data Management in India"

_technologies, doi:10.3390/technologies11010005_

Round 1

Reviewer 1 Report

A recent development in the Internet of Things (IoT) has accelerated the application of IoT-based solutions in healthcare. Next-Gen networks and IoT, supported by the development of technologies such as Artificial Intelligence (AI) and blockchain, have propelled the growth of e-health applications. However, there are some unique challenges in the widespread acceptance of IoT in healthcare. Safe storage, transfer, authorized access control, and the privacy and security aspects of patient data management are crucial barriers to the widespread adoption of IoT in healthcare. This makes it necessary to identify current issues in the various health data management systems to develop novel healthcare solutions.

The authors, in a case study, consider a scheme launched by the Government of India for tuberculosis care called Nikshay Poshan Yojana (NPY).

 It is a web-based Direct Benefit Transfer scheme to provide nutritional incentive of INR 500/- per month to all tuberculosis patients.

The main objective of the authors is to identify the current implementation challenges of the NPY scheme from patient and healthcare stakeholder perspectives and proposes a blockchain-based architecture called NikshayChain for sharing patient medical reports and bank details among several healthcare stakeholders within or across Indian cities.

This architecture can significantly improve medical care, incentive transfer, and data verification.

Their  proposed architecture seems to accelerate healthcare stakeholder productivity by reducing workload and overall costs while ensuring effective data management, propelling the use of e-health applications.

The study is interesting.

However, it need some improvements.

I have some comments.

1.     Rewrite the abstract better proportionally summarizing the sections. For example the aims are near to the end of the abstract.

2.     “In summary, the main contribution of this work is: 142

• Discussion and analysis of the working of Nikshay Poshan Yojana, the responsibili- 143

ties and challenges faced by different stakeholders, and the data management issues 144

associated with this novel scheme. 145

• Proposing a method based on blockchain for smart healthcare and understanding 146

how a blockchain-based architecture can help build patient-centric and efficient data- 147

sharing with a specific example of the TB scheme. 148

• Proposing the architecture for implementing the blockchain-based model for the 149

NPY scheme, which can be extended to any public health scheme. 150

The rest of the paper is organized as follows:-------------“. Please divide the aim from the summary of the design of the paper

3.     Describe figure 1 in details.

4.     Avoid the use of “our”

5.     All the figure must be cited and described in the body of the manuscript

6.     Are section 3 and 4 both results? If yes they must be combined.

7.     The discussion does not contain comparison to the literature

Reviewer 2 Report

This manuscript demonstrated the use of blockchain technology for TB data management systems. The manuscript must be enhanced with the following comments:

1) Add one or two sentences about challenges of the study and later discuss about the contributions made.

2) Figure 1 should include system design which is very important part. 

3) System Design should be added in section 3, with clear steps

4) Figures are not clear (e.g. Figure 5), Visibility of figures should be clearer.

5) The implementation and results discussion, should be change to be Case Study, because you have implement any things as you have mentioned in page 3 , line 135 "It is still in the initial development phase." 

Round 2

Reviewer 1 Report

N/A